# Compositional Analysis of Four Kinds of Citrus Fruits with an NMR-Based Method for Understanding Nutritional Value and Rational Utilization: From Pericarp to Juice

**DOI:** 10.3390/molecules27082579

**Published:** 2022-04-16

**Authors:** Yong Pei, Chenxi He, Huili Liu, Guiping Shen, Jianghua Feng

**Affiliations:** 1Department of Electronic Science, Fujian Provincial Key Laboratory of Plasma and Magnetic Resonance, Xiamen University, Xiamen 361005, China; 33320201150291@stu.xmu.edu.cn (Y.P.); hechenxi@foxmail.com (C.H.); gpshen@xmu.edu.cn (G.S.); 2State Key Laboratory of Magnetic Resonance and Atomic and Molecular Physics, Innovation Academy for Precision Measurement Science and Technology, Chinese Academy of Sciences, Wuhan 430071, China; liuhuili@wipm.ac.cn

**Keywords:** citrus fruit, juice and pericarp, nuclear magnetic resonance, origin identification, nutritional analysis

## Abstract

Citrus is one of the most important economic crops and is widely distributed across the monsoon region. Citrus fruits are deeply loved by consumers because of their special color, fragrance and high nutritional value. However, their health benefits have not been fully understood, especially the pericarps of citrus fruits which have barely been utilized due to their unknown chemical composition. In the present study, the pericarp and juices of four typical varieties of citrus fruits (lemon, dekopon, sweet orange and pomelo) were analyzed by NMR spectroscopy combined with pattern recognition. A total of 62 components from the citrus juices and 87 components from the citrus pericarps were identified and quantified, respectively. The different varieties of the citrus fruits could be distinguished from the others, and the chemical markers in each citrus juice and pericarp were identified by a combination of univariate and multivariate statistical analyses. The nutritional analysis of citrus juices offers favorable diet recommendations for human consumption and data guidance for their potential medical use, and the nutritional analysis of citrus pericarps provides a data reference for the subsequent comprehensive utilization of citrus fruits. Our results not only provide an important reference for the potential nutritional and medical values of citrus fruits but also provide a feasible platform for the traceability analysis, adulteration identification and chemical composition analysis of other fruits.

## 1. Introduction

Citrus is one of the main economic fruit crops, which is widely popular around the world because of its flavor, aroma and taste, high nutritional value and its many health benefits. Citrus fruits can be supplied locally in more than 140 countries across the world, and their annual output has achieved 82 million tons and rapid growth in the past few decades. Besides the pleasing organoleptic characteristics and special color, a variety of active components in citruses, such as carotenoids, flavonoids and flavanones, provide many favorite health-related properties including antioxidant, anti-aging, anti-cancer, antimicrobe, and health-protective properties [1].

The quality, sensory properties and nutritional value of citrus fruit are of great importance to the consumers and the traders in both the domestic and international markets [2]. These characteristics are closely related to the inherent chemical diversity derived from the cultivated varieties and the specific geographical origins. Therefore, it is necessary to develop simple and reliable methods to determine the chemical fingerprints related to specific varieties and comparatively analyze the nutritional components in citrus fruits. Fresh and processed pulp and juice are the most commonly used edible forms of citrus, however, the consumption of citrus pulp or the production of citrus juice usually produces a large amount of residue [3]. The world produces a lot of citrus peels every year, and what to do with it becomes a challenging problem. In China, a small amount of citrus peel is used in traditional Chinese medicine formulations, but almost 99% of the pericarp is discarded as waste. Generally, these discarded materials are no longer used, but they actually contain a large amount of fibers, protein, polyphenol, and essential oils and can be used to produce value-added products which are very beneficial to the human body [4]. Currently, citrus by-products are predominantly utilized as animal feed. Extracting and analyzing those nutrients in citrus peel is becoming more and more attractive, however, fully understanding their chemical composition before their comprehensive utilization is needed [5]. If this abandoned citrus peel can be provided as an alternative treatment, it can not only increase the economic benefits but also even provide additional values [6,7]. If the comparative analysis of these nutrients is fulfilled, reasonable dietary guidance could be provided for citrus consumption.

Currently, metabolomic-based methods are being developed to validate food quality and its nutritional benefits. With the aid of all kinds of analytical methods, including liquid chromatography combined mass spectrometry (LC-MS), gas chromatography combined mass spectrometry (GC-MS), capillary electrophoresis, infrared spectroscopy and nuclear magnetic resonance (NMR) [8,9,10], the changes and variations in food composition can be effectively compared and analyzed by the developed metabolomics methods. Wang et al. [11] investigated the natural variations and spatiotemporal distribution of nutritional components, including amino acids, flavonoids and limonin, in the four fruit tissues (flavedo, albedo, segment membrane and juice sacs) of different citrus species (lemon, pummelo and grapefruit, sweet orange and mandarin) by LC-MS-based metabolomic analysis. Their study revealed differential accumulation patterns of both primary and secondary metabolites in various tissues of the different citrus species. Cluster analysis based on levels of the metabolites detected in the citrus fruits clearly divided the species into four distinct clusters: pummelos and grapefruits, lemons, sweet oranges and mandarins.

The excellent repeatability and reproducibility of NMR technology make it more advantageous than other analytical methods. Different from other analytical techniques, NMR can accurately identify and quantify multiple chemical components simultaneously with high selectivity and without complex chemical pretreatment [12,13]. Moreover, NMR is a non-destructive, high-throughput and detection-rapid analytical tool. A combination of NMR and multivariate statistical analyses, such as principal component analysis (PCA) and orthogonal partial least squares discriminant analysis (OPLS-DA), can visualize the distribution of the samples and organize complex datasets according to their inherent similarity or composition. A non-targeted NMR method was used to simultaneously identify primary and secondary metabolites in polar extracts of grapes, oranges, apples, oranges, bananas, kiwifruit, mangoes, cucumbers, black raspberries, melons, watermelons, blueberries and peaches [14,15,16], and the results indicated that the NMR-based metabolomics method is well suited for establishing specific changes in the chemical characteristics relating to the geographical origin and nutritional quality of these plants and their cultivated varieties. NMR techniques can analyze dozens of components and their changes between different samples in a few minutes/hours with high accuracy and low cost. For example, Eisenmann et al. [14] studied apple peel and pulp extracts of various apple varieties in the different or newly cultivated markets by NMR technology to assess whether it is suitable for distinguishing different varieties. They found that multivariate analysis of NMR data of the pericarp and pulp extracts was able to clearly identify all varieties, while the pericarp extracts showed higher discriminative power. PCA combined with low field ^1^H NMR spectrum also provided good discrimination of five mango cultivars including Awin, Carabao, Keitt, Kent and Nam Dok Mai [17] and 2D NMR gave great help in the identification of various components in mango juice. There are some reports [2,18,19] on the NMR profiling and chemical composition of citrus species produced in the different geographical origins including China. However, there is a lack of detailed chemical composition of the citrus pericarp.

In this study, NMR technology combined with multivariate statistical analyses was applied to comparatively analyze the compositional differences of four kinds of citrus species including lemon, dekopon, sweet orange and pomelo, and the chemical components in the pericarp and juice were quantified. The aim of this study was to provide diet nutrition recommendations and understand the chemical compositions of citrus pericarp for further utilization.

## 2. Materials and Methods

### 2.1. Citrus Sample Collection and Preparation

Four species of citrus fruits were purchased from their main geographical origins in China. Among them, 10 sweet orange (*Citrus sinensis*) samples were from Meishan City, Sichuan Province, 10 lemon (*Citrus limon*) samples were from Ziyang City, Sichuan Province, 10 pomelo (*Citrus maxima*) samples were from Luzhou City, Sichuan Province, and 10 dekopon (*Citrus reticulate Siranui*) samples were from Chengdu City, Sichuan Province of China.

Preparation of citrus juice samples [20]: All the citrus fruits were squeezed, respectively, into a 15 mL Eppendorf (EP) tube using a commercial benchtop juicer. The juice from each citrus fruit (2 mL) was centrifuged at 10,000 g and 4 °C for 10 min, and 400 μL of supernatant was then taken out and mixed well with 200 μL of deuterated phosphate buffer solution (300 mM, pH = 3.82) containing 0.05% TSP (sodium 3-(trimethylsilyl)-2,2,3,3-*d*_4_ propionate) with a vortex instrument. The mixture was allowed to stand for 5 min at room temperature and centrifuged at 10,000× *g* at 4 °C for 10 min. Subsequently, 550 μL of supernatant was transferred into a 5 mm NMR tube (ST500, NORELL, Inc., Morganton, NC, USA) for NMR spectral acquisition. Finally, 40 juice samples composed of 10 biological replicates from each variety were obtained.

Preparation of citrus pericarp samples: pre-weighted 300 mg pericarp tissue from each citrus fruit was put into a 5 mL EP tube with two grinding ceramic beads. The pericarp was ground in a tissue grinder in the form of grinding (30 s)–waiting (15 s) for 3 times at a frequency of 70 Hz. The pericarp slurry was homogenized with 1 mL of methanol, 2 mL of chloroform, and 1 mL of distilled water and vortexed for 60 s. After 10 min partition on ice, the mixture was centrifuged for 10 min at 10,000× *g* at 4 °C. The supernatant was transferred into a 1.5 mL EP tube and lyophilized for 24 h to remove methanol and water. The freeze-dried pericarp powder was dissolved in 600 μL of deuterated phosphate buffer solution (200 mM, pH = 6.5) containing 0.05% TSP. After vortex mixing for 5 min, the extract was subjected to centrifugation for 10 min at 4 °C for 10,000× *g*. Subsequently, 550 μL of supernatant was transferred into a 5 mm NMR tube for NMR spectral acquisition. Finally, 40 pericarp samples composed of 10 biological replicates from each variety were obtained.

### 2.2. ^1^H-NMR Spectroscopy and Spectral Preprocessing

The NMR measurements of all of the citrus juice and pericarp samples were performed on an 850 MHz Bruker Avance III NMR spectrometer (Bruker Corporation, Kalsruhe, Germany) equipped with a CPTCI probe operating 850.29 Hz and under the room temperature of 298 K. All the ^1^H-NMR spectra were acquired using a standard ZGPR pulse sequence. For each sample, 64 FIDs (free induction decays) were collected into 32 K data points over a spectral width of 17 KHz with a relaxation delay of 4 s and an acquisition time of 1.93 s.

The NMR spectra were preprocessed with MestReNova software (V9.0.1, Mestrelab Research, Santiago de Compostela, Galicia, Spain). For all of the ^1^H-NMR spectra, FIDs were multiplied by an exponential function with a 0.3 Hz line-broadening factor to improve the signal-to-noise ratio before Fourier transform. The NMR spectra were then manually phased, baseline-corrected (polynomial fitting) and calibrated to the singlet of TSP at δ 0.0. The spectral regions of δ 0.2–10.0 were automatically integrated with the integral interval of 0.005 ppm, and the residual water signals (δ 4.65–5.20 for citrus juice samples and δ 4.81–4.89 for citrus pericarp samples) and methanol signals (δ 3.35–3.37 for citrus pericarp samples) were removed in order to avoid the interference of the residual methanol and water. The integrated data were finally normalized to the total sum of the spectrum.

In order to confirm the assignment of the NMR signals of the citrus fruits, a series of 2D-NMR spectra, including ^1^H-^1^H COSY, ^1^H-^1^H TOCSY, ^1^H-^13^C HSQC, ^1^H-^13^C HMBC and J-resolved NMR spectra, were also acquired on the selected citrus juice and pericarp samples.

### 2.3. Univariate and Multivariate Statistical Analyses

The obtained NMR integral data were imported into SIMCA 14.1 (Umetrics, Umea, Sweden) for the multivariate statistical analyses including principal component analysis (PCA), partial least squares discriminant analysis (PLS-DA) and orthogonal partial least squares discriminant analysis (OPLS-DA). Pareto scaling was chosen to display the contribution from both the major and minor components. In this study, the pair-wise OPLS-DA models between one of the citrus fruit juices/pericarp and three other citrus varieties fruit juices/pericarp were established to identify the most significant chemical markers in each citrus fruit. The cross-validation analysis of variance (Cross Validation Analysis Of Variance, CV-ANOVA) evaluated the significance of the OPLS-DA model by calculating the *p*-value, and permutation test (permutation number = 300) was used to verify whether the model was over-fitted.

Finally, a four-dimensional volcano map was used to identify the potential chemical markers, which play a great role in the separation of the pair-wise groups. In the map, fold change is defined as the concentration ratio of a substance in two groups of juice or pericarp, p-value was obtained by the Student’s t-test of the concentration of the substance in two groups. In our study, log_2_(fold change) and −log_10_(*p*-value) represent the x- and y-axes of volcano map, respectively. The absolute correlation coefficient (│r│) and the variable importance for the projection (VIP) value of each component from the OPLS-DA model are represented in the volcano plot by the circle size and the color, respectively. The potential chemical markers were screened out by three criteria: *p* < 0.05, │r│ > 0.500, and the VIP value in the top 10%. The volcano maps are generated by MATLAB (R2014b, download from http://www.mathworks.com) with self-compiled scripts.

In addition, each component in the citrus juice and pericarp was quantified by comparing the integral of the characteristic peak with that of the internal standard (TSP) according to our previous report [20].
$$C_{x}=\frac{A_{x}}{N_{x}}\times \frac{N_{TSP}}{A_{TSP}}\times C_{TSP}$$
where *C_x_* is the molar concentration of any a component x in the juice or pericarp samples, in units of mol/L; *A_x_* is the integral area of this component in the ^1^H-NMR spectrum after correction of spin-lattice relaxation time (T_1_); *N_x_* is the number of protons contributing to the NMR signals; *C_TSP_* is the molar concentration of TSP in the juice or pericarp samples, in units of mol/L; *A_TSP_* is the integral area of TSP at δ 0.00 in the same ^1^H-NMR spectrum after T_1_ correction; *N_TSP_* is the number of protons contributing to the singlet at δ 0.00 (here, it is 9). The results were represented as mean ± standard deviation (SD).

## 3. Results and Discussion

### 3.1. The Comparison of ^1^H-NMR Spectral Profiles and Chemical Composition of Different Varieties of Citrus Juice and Pericarp

A total of 80 samples composed of 10 biological replicates from each species of citrus juice and pericarp were detected by the ^1^H-NMR technique. The superimposed ^1^H-NMR spectra of the four species of citrus juices and pericarps are displayed in Figure 1A,B, respectively. The signals in the spectra were assigned to the individual components according to the related literature [2,10,20,21,22,23,24] and the 2D-NMR spectra of the selected juice and pericarp samples, including ^1^H-^1^H COSY, ^1^H-^1^H TOCSY, ^1^H-^13^C HSQC, ^1^H-^13^C HMBC and J-resolved NMR spectra, were confirmed by the in-house developed NMR database based on etalon compounds and public NMR databases such as HMDB [25] and BMRB [26]. The components identified from these citrus fruits were verified by other techniques including LC-MS [27,28,29] or GC-MS [30,31] in previous studies (Table 1). The NMR signals of some components could be confirmed using similar studies [32,33,34,35] by the NMR technique (Table 1). A total of 102 components were identified from the NMR spectra of the citrus fruits, among which, 15 components are juice-specific and 40 components are pericarp-specific, and the rest of the components are shared by the juices and pericarps. The detailed spectral information of the citrus juice and pericarp was tabulated in Table 1.

The NMR spectra profiles give a summary and comparison of the chemical components in the four varieties of citrus fruits. In general, the four citrus fruit juices display quite similar chemical profiles. The carbohydrate regions (δ 3.0–5.5) dominate the NMR spectra both of the juice and pericarp, in which sucrose, α- and β-glucose and fructose demonstrate the higher concentrations, and the other low-levels of carbohydrates, including erlose, fucose, maltotriose, melezitose, raffinose, rhamnose and trehalose, are also observed in this region. Sugar accumulation in citrus fruits is a species-dependent response. These carbohydrates contribute to the sweet taste of citrus fruit juice and also play important roles in fruit maturation and ripening [10]. In the aromatic regions (δ 5.5–10.0), some minor components were identified from the juices and pericarps including the phenolic compounds such as flavonols (naringenin), flavonoids (hesperidin, hesperetin, flavone, phenindione), flavanone (naringin, neohesperidin), 4-nitrophenol, corticosterone, ellagic acid, limonin and γ-terpinene, and the other aromatic compounds such as 1-methylhistidine, 2-furoic acid, 3-methylphenylacetic acid, linalool, phenylalanine, synephrine, sphingosine and trigonelline. The signals at the spectral region of δ 0.5–3.0 mainly come from the various amino acids (e.g., alanine, lysine, glutamine, arginine, valine and proline) and organic acids (e.g., citric acid, ethanol and malic acid). Some components will give a larger peak drift with pH, and citric acid is the most typical one. In this study, the pH of the citrus juice was 3.82, while the pH of the citrus pericarps was 6.5. The variation of the pH led to the changes in the chemical shift of citric acid from δ 2.75 and δ 2.85 (at the spectra of juice) to δ 2.55 and δ 2.55 and δ 2.73 (at the spectra of pericarp), as shown in Figure 1 and Table 1. It is not surprising that the content of citric acid in lemon juice is much higher than in the other citrus juices. Citric acid contributes to the sour and astringent taste of lemon juice [10], and a higher level of citric acid was also observed in lemon pericarp than the other citrus pericarp. It is clear from the spectra that the minor components in the citrus fruits would provide the differential compositions between the species though the main components share similar profiles.

In this study, the concentration of each component in the juice and pericarp from each citrus fruit was quantified by comparing the integral of the characteristic peak with that of the internal standard (TSP) [20]. Finally, 62 components in the citrus juices and 87 components in the citrus pericarps were quantificationally obtained, and the results were summarized in Appendix A.

### 3.2. Compositional Differences between the Citrus Juices in the Different Varieties

The principal component analysis (PCA) can reduce the dimension of input variables, visualize the clustering and separation between the different varieties of citrus juice and further identify the possible outliers. Figure 2A shows the PCA score plots of the four varieties of citrus juices, where lemon juices separate obviously from the other three kinds of citrus juices, while the dekopon, sweet orange and pomelo juices overlap with each other. This is mainly because the composition of lemon juice is obviously different from the other three citrus juices as clearly seen from their spectral analysis. The supervised partial least squares discriminant analysis (PLS-DA) was also performed on all the citrus juices. The sample distribution of the PLS-DA is similar to that of PCA but highlights the inter-group separation and intra-group clustering (Figure 2B). The lemon juice samples were located on the right side of the first principal component (PC1), and the dekopon, sweet orange and pomelo juices could be distinguished along PC2 although there are still some overlaps between the dekopon juices and the pomelo juices, which may be related to the proximity of their juice composition.

To screen out the most significant differential components in each citrus juice, four OPLS-DA models were established on the pair-wise sample groups including lemon vs. the other three citrus juices, dekopon vs. the other three citrus juices, sweet orange vs. the other three citrus juices and pomelo vs. the other three citrus juices. Figure 3 displays the OPLS-DA score plots (left panels) and the corresponding volcano plots (right panels) to screen out the differential components in each citrus juice. The model parameters including R^2^X, R^2^Y, Q^2^ and the *p*-value from CV-ANOVA were attached to the figures, and the detailed statistical analysis of the differential components was listed in Appendix A, including the p-value and fold change from the univariate analysis and the correlation coefficient and VIP value from OPLS-DA.

Figure 3A shows the OPLS-DA model of lemon juice vs. the other three citrus juices. The separation between the two groups of samples was obvious and the Q^2^ reached an astonishing 0.988, while the R^2^Y was also a higher value (0.993). The p-value from cross-validation ANOVA of the model was only 8.69 × 10^−35^, and the results of the permutation test indicated no over-fitting, which further confirmed the obvious compositional difference between the lemon juice and the other three citrus juices. With the selection criteria, eleven significantly differential components were screened out from the volcano plot of the lemon juice including higher levels of betaine, citric acid, ethanol, glutamine, malonic acid and *scyllo*-inositol, and lower levels of fructose, glycine, sucrose, α- and β-glucose (Appendix A). The accumulation of citric acid and ethanol can be affected by the maturation and ripening stages, and they were recognized as a determining factor in fruit quality [10,20]. The higher level of ethanol could also be because of the possible fermentation of lemon juice during storage and transportation [35]. This was supported by the decreased levels of sugars including fructose, α- and β-glucose. The combination of sugar and organic acids including citric acid and malonic acid could play an important role in rating the organoleptic property of the fruit [10]. Citric acid has low toxicity and is widely used in the pharmaceutical and food industries, and it can also be used in cleaning products and cosmetics. Therefore, the extremely high content of citric acid in lemon juice enables its broad application. *Scyllo*-inositol is also quite rich in lemon juice. Alzheimer’s disease involves the aggregation of amyloid β proteins, causing neuronal apoptosis and loss of cognitive function. As a rare stereoisomer of inositol, *scyllo*-inositol can reach the brain and prevent amyloid β proteins from forming toxic amyloid fibrils and polymers, thus providing a promising therapeutic for Alzheimer’s disease [36]. The high content of *scyllo*-inositol in lemon juice may provide a new idea for the efficient comprehensive utilization of lemon.

Figure 3B shows the OPLS-DA model of dekopon juice vs. the other three citrus juices with a Q^2^ of 0.871 and an R^2^Y of 0.900. The permutation test with 300 permutations and the p-value from CV-ANOVA (8.10 × 10^−13^) indicates the obvious compositional differences between dekopon juice and the other citrus juices. Five significant chemical markers were screened out from dekopon juice according to the corresponding volcano plot including higher levels of glycine, proline and sucrose and lower levels of citric acid and α-glucose (Appendix A). Proline is not only an important part of protein but also plays an important role in adapting to osmotic and dehydration stress, redox control and apoptosis [37]. Further research on proline may provide effective methods for the full utilization of dekopon. The content of glycine in dekopon juice is also relatively high, and some research results show that glycine strengthens the regulation of dietary amino acid levels, thereby prolonging the healthy life of mice, and laying the foundation for further research on the effects of diet on aging and diseases in later life [38]. This undoubtedly opens up whole new vistas for the study of dekopon.

The OPLS-DA model of sweet orange juice vs. the other three citrus juices is shown in Figure 3C. From the score plots, the separation between the two groups is obvious, the Q^2^ is 0.932, the R^2^Y is 0.942 and the *p*-value from CV-ANOVA is only 8.10 × 10^−13^. The permutation test further confirmed that there are obvious differences between the sweet orange juice and the other citrus juices. Eight differential components were screened out from the sweet orange juice including higher levels of acetic acid, fructose, glycine, malic acid, proline, α- and β-glucose and lower levels of citric acid (Appendix A). Malic acid gives a smooth taste that disappears gradually in the mouth. The combination of higher malic acid and lower citric acid with sugars such as fructose and glucose and amino acids such as glycine and proline results in a sweet–sour taste and could play an important role in rating the organoleptic property and quality of the citrus fruits [10]. Acetic acid, the main acid component of vinegar, is often used to determine the antiplatelet and fibrinolytic activity, while organic acids play an important role in the prevention and treatment of cardiovascular diseases [39].

Figure 3D displays the OPLS-DA model of pomelo juice vs. the other three citrus juices with the Q^2^ of 0.825, the R^2^Y of 0.853, the p-value from CV-ANOVA of 1.80 × 10^−22^, and the results of the permutation test further confirmed the obvious differences between the two groups. Nine significantly differential substances were screened out from pomelo juice based on the screening criteria including higher levels of glycine, *N*,*N*-dimethylglycine (DMG) and sucrose and lower levels of citric acid, ethanol, malonic acid, methanol, *scyllo*-inositol and trimethylamine N-oxides (TMAO) (Appendix A). As can be seen from the volcano plots, the content of DMG is much higher in the pomelo juice than in the others. Betaine (*N*,*N*,*N*-trimethylglycine) forms DMG in mitochondria, then further demethylates to myosine (*N*-methylglycine), and finally to glycine, and this is consistent with the higher glycine content in pomelo juice. It can also be seen from the volcano plot that the content of sucrose is also very high, so it is recommended that people with hyperglycemia or diabetes should not consume it much.

### 3.3. Compositional Differences between the Citrus Pericarps in the Different Varieties

Figure 2C shows the PCA score plots of the four varieties of citrus pericarps, in which the four kinds of citrus peel samples are basically dispersed in three areas, and only the dekopon peels and sweet orange peels overlapped more with each other. Their similarity could be confirmed from their NMR spectra, where the difference between these two kinds of pericarp is quite small, which may be related to the similarity in their chemical compositions. One pomelo peel sample appears outside the T_2_ Hotelling ellipse (95% confidence interval). According to the analysis of the corresponding original NMR spectra, no obvious spectral abnormality was observed between this sample and the others. Therefore, this sample was not identified as an outlier but as an individual with differences during pomelo growth. The distribution profile of the citrus pericarps in PLS-DA is similar to PCA, but PLSL-DA highlighted the differences between the different varieties and the intra-group clustering (Figure 2D). There is still a small overlap between the dekopon and sweet orange peels, which is consistent with the PCA results.

To identify the significantly differential components in each citrus pericarp, four OPLS-DA models were established: the lemon vs. the other three citrus pericarps, the dekopon vs. the other three citrus pericarps, the sweet orange vs. the other three citrus pericarps and the pomelo vs. the other three citrus pericarps. Figure 4 displays the OPLS-DA score plots (left panels) and the corresponding volcano plots (right panels) to screen out the differential components in each citrus pericarp. The model parameters including R^2^X, R^2^Y, Q^2^ and the p-value from CV-ANOVA were attached to the figures, and the detailed statistical analysis of the differential components was listed in Appendix A, including the p-value and fold change from the univariate analysis and the correlation coefficient and VIP value from OPLS-DA.

Figure 4A shows the OPLS-DA model of lemon pericarp vs. the other three citrus pericarps. The separation between the two groups of samples is obvious with favorable model parameters including 0.743 of the Q^2^, 0.873 of the R^2^Y, and only 6.81 × 10^−10^ of the *p*-value from CV-ANOVA, which further indicated the obvious compositional differences between the lemon and the other three citrus pericarps. Fifteen differential components were screened out from lemon peel including higher levels of asparagine and citric acid and lower levels of betaine, cholic acid, fructose, galactose alcohol, glycerol, malonic acid, inositol, neopterin, o-hydroxybenzoic acid, sucrose, taurine, α-glucose and β-glucose (Appendix A). As seen from the volcano map, most of these differential components are relatively lower in the lemon peel than in the other citrus peels, however, the contents of two components, citric acid and asparagine, are higher in the lemon peel than in other citrus peels. Studies have shown that asparagine can regulate blood pressure, and the high levels of asparagine in lemon peel may provide new ideas for the role of lemon peel in the field of drugs [40].

Figure 4B shows the OPLS-DA model of the dekopon pericarp vs. the other three citrus pericarps with a Q^2^ of 0.677, an R^2^Y of 0.713, and the p-value from CV-ANOVA of 3.40 × 10^−8^, which indicate the obvious differences between the dekopon peel and the other citrus peels. Nine significantly differential components were screened out from dekopon peel including higher levels of ethanolamine, fructose, glutamic acid, glycerol and neopterin and lower levels of hesperidin, methanol, quinic acid, and rhamnose (Appendix A). In recent years, new physiological functions of neopterin have been discovered, such as inducing or enhancing cytotoxicity, inducing apoptosis and its role as a chain-breaking antioxidant [41]. Glutamic acid has various important functions such as endogenous anticancer agents, conjugates of anticancer agents, and glutamic acid derivatives as possible anticancer agents [42]. The research on glutamic acid may be beneficial to the full use of dekopon peel and for avoiding the loss of nutritional value.

Figure 4C shows the OPLS-DA model of sweet orange pericarp vs. the other three citrus peels. The model parameters including the Q^2^ of 0.508, the R^2^Y of 0.614, and the p-value from CV-ANOVA of 4.09 × 10^−5^ indicate that there are obvious compositional differences between the sweet orange peel and the other citrus peels, while the results of the permutation test show that the model has not been over-fitted. Only three significant differential substances were screened out from the sweet orange peel: alanine, galactoseol and succinimide, and all of their contents are higher in the sweet orange peel than in the other citrus peels. Alanine can prevent kidney stones, assist glucose metabolism, help alleviate hypoglycemia, improve body energy and other medicinal effects, which may provide a new idea for the use of sweet orange peel. Succinimides are well recognized heterocyclic compounds in drug discovery that produce diverse therapeutically related applications in pharmacological practices [43]. Moreover, its derivatives have a variety of medicinal properties, such as anticonvulsants, anti-inflammatory drugs, antitumor drugs and antibacterial drugs. If the succinimide in the orange peel can be extracted to obtain its derivatives, the nutritional value of the orange peel will be further expanded.

Figure 4D shows the OPLS-DA model of the pomelo peel vs. the other three citrus peels. The Q^2^ is 0.848, the R^2^Y is 0.879, and the p-value from CV-ANOVA is 7.68 × 10^−14^, indicating that there are obvious compositional differences between the two groups of samples, and the results of the permutation test confirmed no over-fitting of the model. Eight significantly differential substances including higher levels of betaine, choline, hesperidin, malonic acid, methanol, quinic acid and rhamnose and lower level of glutamic acid were screened out from the pomelo pericarp (Appendix A). Studies have shown that phenolic compounds, being an important source of flavanones, especially hesperidin, also known as vitamin P, have a variety of biological properties such as anti-inflammatory, anti-cancer, antioxidant effects and reducing capillary fragility and lowering blood lipid activity [20,44]. The astonishing high level in the pomelo peel suggests that the extraction and comprehensive utilization of pomelo peel are more economical than the other three pericarps. The content of methanol and rhamnose is also high, and rhamnose can be used to determine the permeability of the intestinal tract and can also be used as a sweetener or produce flavors and fragrances.

## 4. Conclusions

In this study, the chemical composition of four varieties of citrus fruits, including lemon, dekopon, sweet orange and pomelo, were analyzed and compared by NMR spectroscopy combined with stoichiometry techniques. The main and minor components in the citrus juices and pericarps were identified and quantified by NMR spectroscopy. A total of 62 components were identified from the citrus juices and 87 components from the citrus pericarps. According to our screening criteria, the corresponding chemical markers, especially the higher components, were selected from each citrus fruit juice and pericarp, which contribute to the daily diet recommendation and the understanding of their rational utilization. The chemical markers revealed that lemon presents high fruit quality and maturation status, and both its juice and pericarp have high antioxidant potential, therefore giving it potential in the pharmaceutical and food industries. Dekopon displays some healthy factors in adapting to dehydration stress, redox control and apoptosis. Sweet orange presents favorable organoleptic properties and high fruit quality, and its consumption in the daily diet is also preferred for the prevention of cardiovascular and other diseases. Although pomelo juice is not suitable for hyperglycemia and diabetes patients, the high content of flavonoids, especially hesperidin, in pomelo peel may bring new economic benefits. Our results provide the daily diet recommendation of the four varieties of citrus juices and the basis for the rational and comprehensive utilization of citrus pericarp.

## Figures and Tables

**Figure 1 molecules-27-02579-f001:**
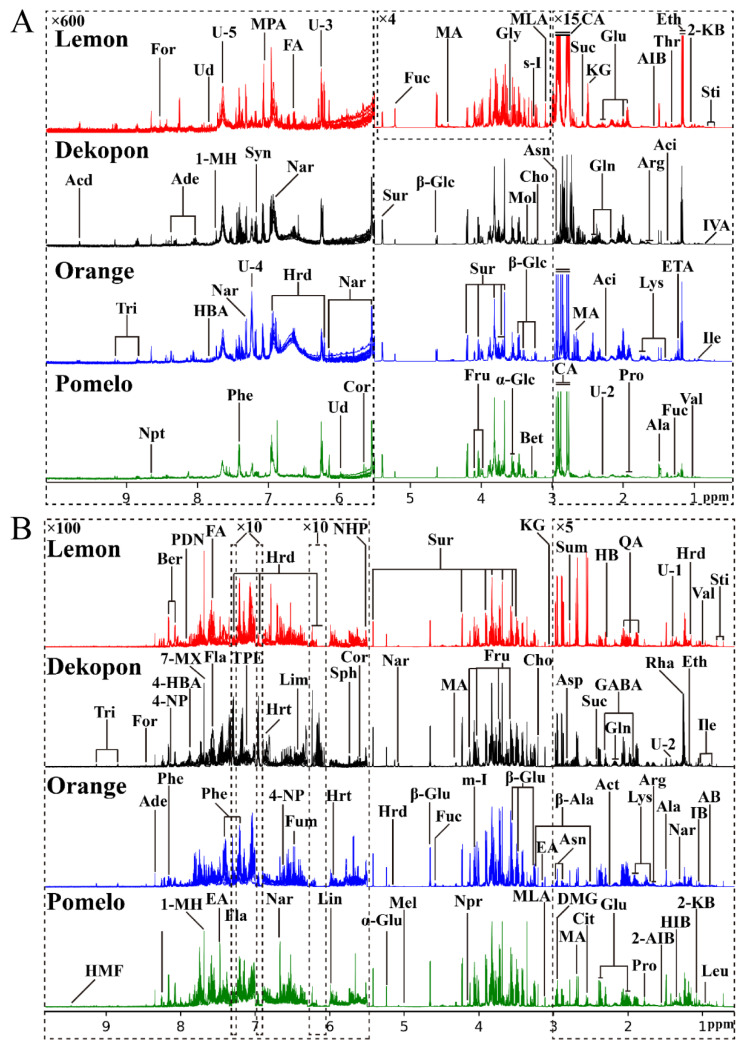
The superimposed 850 MHz ^1^H-NMR spectra of the four varieties of citrus juices (**A**) and pericarps (**B**). For clarity, the spectral regions of δ 0.5–3.0 were vertically magnified 15 times and 5 times relative to the spectral regions of δ 3.0–5.5 for the juices and pericarps, respectively. The spectral regions of δ 3.0–5.5 of the lemon juice were vertically magnified 4 times relative to the same spectral region of the other three citrus juices. The spectral regions of δ 5.5–10.0 were vertically magnified 600 times and 100 times relative to the spectral regions of δ 3.0–5.5 for the juices and pericarps, respectively, among which the spectral regions of δ 6.00–6.30, δ 6.90–7.00 and δ 7.31–7.35 for the pericarps were only magnified 10 times. The keys and the detailed spectral information of the identified components are shown in Table 1. These U on the spectra indicate the unassigned signals.

**Figure 2 molecules-27-02579-f002:**
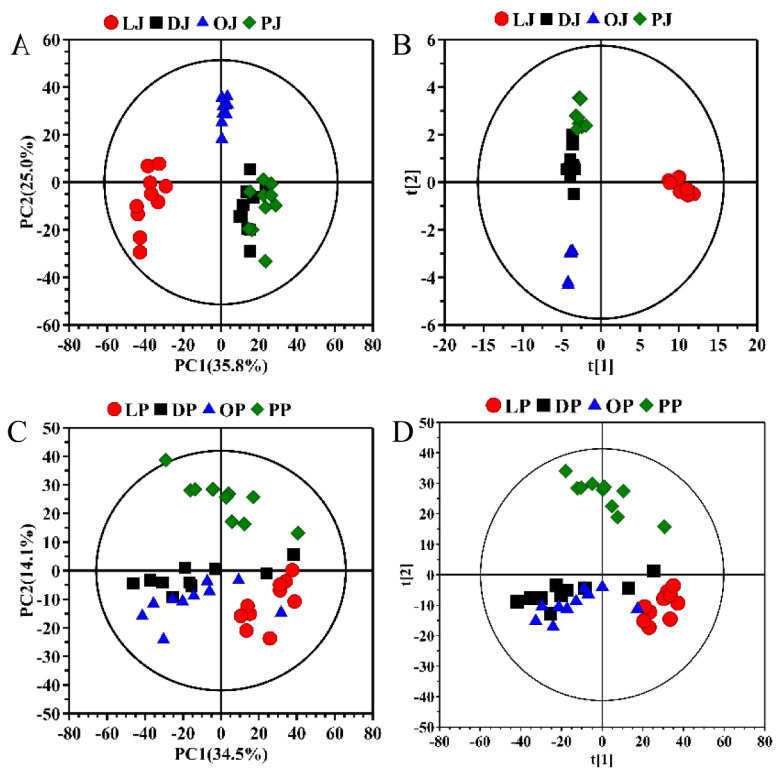
Two-dimensional PCA (**A**,**C**) and PLS-DA (**B**,**D**) score plots of different citrus juices (**A**,**B**) and pericarps (**C**,**D**). LJ: lemon juice; DJ: dekopon juice; OJ: sweet orange juice; PJ: pomelo juice; LP: lemon pericarp; DP: dekopon pericarp; OP: sweet orange pericarp; PP: pomelo pericarp.

**Figure 3 molecules-27-02579-f003:**
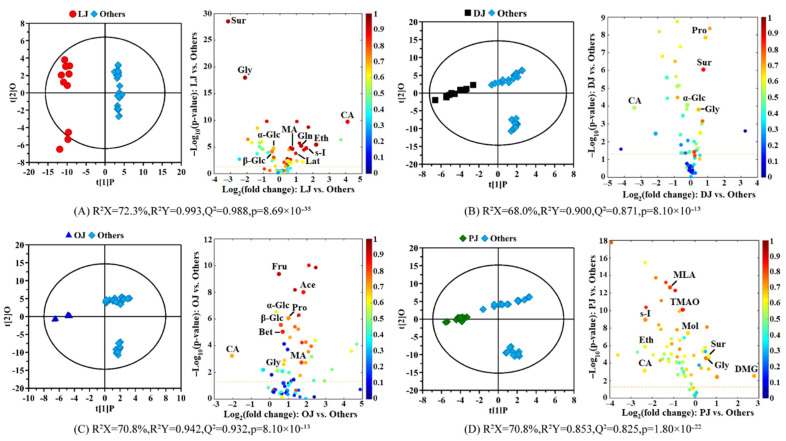
OPLS-DA score plots (left panels) and the corresponding volcano plots (right panels) derived from the NMR data of citrus juices in the different varieties. (**A**) Lemon juice vs. other juices; (**B**) Dekopon juice vs. other juices; (**C**) Sweet orange juice vs. other juices; (**D**) Pomelo juice vs. other juices. LJ: lemon juice; DJ: dekopon juice; OJ: sweet orange juice; PJ: pomelo juice; Others: other three varieties of citrus juices. The marked dots in the volcano plots represent the components with statistically significant differences between the pair-wise groups. The keys of the components were shown in Table 1.

**Figure 4 molecules-27-02579-f004:**
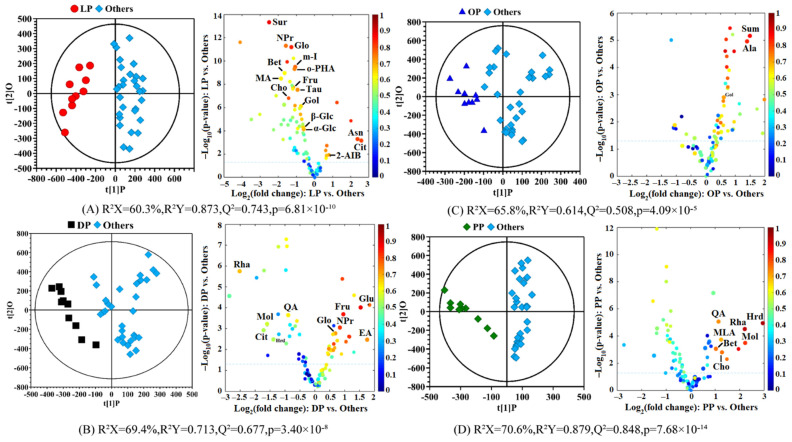
OPLS-DA score plots (left panels) and the corresponding volcano plots (right panels) derived from the NMR data of citrus pericarps in the different varieties. (**A**) Lemon pericarp vs. other pericarps; (**B**) Dekopon pericarp vs. other pericarps; (**C**) Sweet orange pericarp vs. other pericarps; (**D**) Pomelo pericarp vs. other pericarps. LP: lemon pericarp; DP: dekopon pericarp; OP: sweet orange pericarp; PP: pomelo pericarp; Others: other three varieties of citrus pericarps. The marked dots in the volcano plots represent the components with statistically significant differences between the pair-wise groups. The keys of the components were shown in Table 1.

**Table 1 molecules-27-02579-t001:** The identified components from the NMR spectra of juices and pericarps of four varieties of citrus.

Components	Abbr.	Chemical Shift (ppm) (Multiplicity)	Source	Ref.
1-Methylhistidine	1-MH	7.08(s ^a^), 7.73(s)(1) ^b^	Juice	[20,25,26]
1-Methylhistidine	1-MH	7.08(s), 7.68(s)(1)	Pericarp	[20,25,26]
2-Aminoisobutyrate	2-AIB	1.55(s)(6)	Pericarp	[25,26]
2-Furoic acid	FA	6.54(dd), 6.65(d), 7.57(dd)(1)	All ^c^	[20,25,26]
2-Ketobutyric acid	2-KB	1.07(t)(3)	All	[20,25,26]
3-Hydroxybutyric acid	HB	1.21(d)(3), 2.28(dd)	Pericarp	[22,25,26]
3-Methylphenylacetic acid	MPA	7.07(m)(2)	All	[20,25,26]
4-Hydroxybenzoic acid	HBA	7.82(d)(2)	All	[20,25,26]
4-Nitrophenol	4-NP	6.70(m), 8.13(m)(2)	Pericarp	[25,26]
7-Methylxanthine	7-MX	7.93(s)(1)	Pericarp	[25,26]
Acetaldehyde	Acd	2.23(d), 9.67(q)(1)	Juice	[20,25,26]
Acetic acid	Ace	1.91(s)(3)	Juice	[20,21,22,23,24,30,32,33,34,35]
Acetoin	Aci	1.37(d), 2.21(s)(3)	All	[2,20,21]
Acetone	Act	2.25(s)(6)	Pericarp	[25,26]
Adenosine	Ade	6.11(d), 8.09(s), 8.36(s)(1)	All	[20,28,34]
Alanine	Ala	1.47(d)(3)	All	[2,10,20,21,22,23,24,33,34]
Arginine	Arg	1.68(m)(2)	All	[2,20,21,22,24,28,33]
Ascorbic acid	Asc	4.50(d)(1)	Juice	[2,20,33,35]
Asparagine	Asn	2.88(dd), 2.95(dd)(1)	All	[20,22,28,34]
Aspartic acid	Asp	2.67(dd)(1), 2.73(dd)	All	[2,10,20,33,34]
Berberine	Ber	4.91(t), 8.08(d), 8.16(d)(1)	Pericarp	[25,26]
Betaine	Bet	3.27(s)(9), 3.90 (s)	All	[20,22]
Choline	Cho	3.19(s)(9)	All	[2,10,20,21,22,34]
Citric acid	CA	2.75(1/2AB)(2), 2.85(1/2AB)	Juice	[2,10,20,21,22,23,34,35]
Citric acid	Cit	2.55(1/2AB)(2), 2.73(1/2AB)	Pericarp	[2,10,20,22,33,34,35]
Corticosterone	Cor	5.67(d)(1)	All	[20,25,26]
Cuminaldehyde	Cum	9.98(s)(1)	Pericarp	[25,26]
Ellagic acid	EA	7.47(s)(2)	Pericarp	[25,26]
Erlose	Erl	5.36(d)(1)	Juice	[25,26]
Ethanol	Eth	1.17(t)(3), 3.64(q)	All	[2,10,20,21,22,23,32,33,34,35]
Ethanolamine	ELA	3.15(t)(2)	Pericarp	[22,34]
Ethyl acetate	ETA	1.23(t)(3)	Juice	[20,21,30]
Flavone	Fla	7.55(m), 7.62(m), 7.74(m), 7.88(m), 8.24(dd)(1)	Pericarp	[2]
Formic acid	For	8.45(s)(1)	All	[20,23,33,34]
Fructose	Fru	3.58(m), 3.70(m), 3.72(m), 3.80(m), 3.88(dd), 3.99(m), 4.01(m), 4.10(d)(2)	All	[2,10,20,21,22,23,24,33,34,35]
Fucose	Fuc	1.26(t), 4.59(d), 5.26(d)(1)	All	[20,25,26]
Fumaric acid	Fum	6.52(s)(2)	Pericarp	[2,21,23,33,34]
Galactitol	Gol	3.96(t)(1)	Pericarp	[25,26]
Glutamic acid	Glu	2.00(m), 2.07(m), 2.34(m)(2)	All	[20,22,24]
Glutamine	Gln	2.16(m), 2.45(m)(2)	All	[2,20,22,24,28,33,34]
Glycerol	Glo	3.67(dd)(2)	Pericarp	[21,22,32,34,35]
Glycine	Gly	3.57(s)(2)	Juice	[20,25,26]
Gulonolactone	GNA	4.52(s), 4.55(s)(1)	All	[20,25,26]
Hesperedin	Hrd	1.16(m), 3.35(m), 4.52(s), 4.61(m), 4.76(m), 5.13(s), 5.32(dd), 6.16(m), 6.23(d)(2), 6.40(m), 6.80(m), 6.96(d), 7.34(m)	All	[20,27,28]
Hesperitin	Hrt	5.44(q)(1), 5.95(dd), 6.85(m), 6.91(m)	All	[27,28,29]
Histamine	Him	8.00(s)(1)	Pericarp	[25,26]
Hydroxymethylfurfural	HMF	9.47(s)(1)	Pericarp	[25,26]
Isobutyric acid	IB	1.05(d)(6)	Pericarp	[25,26,34]
Isoleucine	Ile	0.93(t), 1.00(d)(3)	All	[2,10,20,21,33,34]
Isovaleric acid	IVA	0.90(t)(6)	All	[20,25,26]
Lactose	Lat	4.46(d)(1),5.17(d)	All	[20,35]
Leucine	Leu	0.95(t)(6)	All	[2,10,20,21,33,34]
Limonin	Lim	6.49(d)(1), 7.53(d),7.71(d)	All	[20,22,28,30,31]
Linalool	Lin	5.97(dd)(1)	Pericarp	[30]
Lysine	Lys	1.42(m), 1.75(m)(2), 1.90(m),3.04(m)	All	[10,22]
Malic acid	MA	2.60(dd), 4.43(dd)(1)	All	[2,10,20,21,22,23,33,34]
Malonic acid	MLA	3.10(s)(2)	All	[2,10,20,34]
Maltotriose	Mal	5.38(m)(1)	Pericarp	[25,26]
Melezitose	Mel	4.99(d)(1)	Pericarp	[25,26]
Methanol	Mol	3.35(s)(3)	All	[2,20,21,22,23,34]
*myo*-Inositol	m-I	3.63(t), 4.08(t)(1)	Pericarp	[2,21,22,34]
N,N-Dimethylglycine	DMG	2.94(s)(6)	All	[20,25,26]
Naringenin	Nan	5.49(d)(2), 7.41(d)	Juice	[20,27,28]
Naringin	Nar	1.28(dd), 5.06(d), 5.54(d), 6.15(t), 6.79(d), 6.94(d), 7.32(d)(2)	All	[20,27,28]
Neohesperidin	NHP	5.51(dd)(1), 5.62(dd)	Pericarp	[29]
Neopterin	NPr	4.13(dd)(1)	Pericarp	[25,26]
Neopterin	NPt	8.65(s)(1)	Juice	[20,25,26]
*ortho*-Hydroxyphenylacetic acid	o-HPA	3.61(s)(2)	Pericarp	[25,26]
*para*-Aminobenzoic acid	p-ABA	6.57(d)(2), 7.66(d)	Pericarp	[25,26]
p-Cresol	Cre	7.12(d)(2)	Juice	[20,25,26]
Phenindione	PDN	7.86(d), 8.04(d)(2)	Pericarp	[25,26]
Phenylalanine	Phe	7.28(m), 7.38(m), 7.45(m)(2)	All	[2,10,20,24,28,33,34]
Proline	Pro	1.94(m), 4.13(dd)(1)	All	[2,10,20,21,22,23,24]
Quinic acid	QA	1.88(dd)(1), 1.97(dt), 2.06(m), 3.94(m), 4.15(q)	Pericarp	[22,23,24]
Raffinose	Raf	4.97(d)(1)	All	[20,25,26]
Rhamnose	Rha	1.20(d), 5.08(d)(1)	All	[20,25,26]
*scyllo*-Inositol	s-I	3.33(s)(6)	Juice	[20,22]
Sebacic acid	Seb	1.31(s)(8)	Pericarp	[25,26]
Sphingosine	Sph	5.72(m)(2)	Pericarp	[25,26]
Stigmasterol	Sti	0.71(s)(6), 0.79(m), 0.81(s), 1.78(m), 5.04(m)	All	[20,25,26]
Succinic acid	Suc	2.63(s)(4)	All	[2,10,20,21,22,23,24,32,35]
Succinimide	Sum	2.78(s)(3)	Pericarp	[25,26]
Sucrose	Sur	3.47(t),3.54(dd), 3.67(s), 3.75(t), 3.81(t), 3.84(m), 3.90(m), 4.05(t), 4.21(d), 5.40(d)(1)	All	[2,10,20,23,24,33,34,35]
Synephrine	Syn	6.89(d), 7.18(d)(2)	Juice	[20,28]
Taurine	Tau	3.28(t)(2)	Pericarp	[25,26]
Threonine	Thr	1.32(d)(3), 4.28(m)	All	[2,10,20,21,22,24,33,34]
Tiglic acid	Tig	1.75(m)(2), 1.76(s)	Pericarp	[25,26]
Trehalose	Tre	5.18(d)(2)	All	[10,20]
Trigonelline	Tri	4.36(s), 8.08(m), 8.83(t), 9.12(s)(1)	All	[2,20,21,22]
Trimethylamine	TMA	2.92(s)(9)	Pericarp	[25,26]
Trimethylamine N-oxide	TMAO	3.29(s)(9)	Juice	[20,25,26]
Uridine	Ud	5.89(d), 5.90(d), 7.87(d)(1)	Juice	[20,34]
Valine	Val	0.98(d), 1.03(d)(3)	All	[2,10,20,21,22,24,33,34]
Vitamin C	VC	4.48(d)(1)	Pericarp	[2,25,26]
α-Amino-N-butyric acid	AB	0.96(t)(3)	Pericarp	[20,25,26]
α-Aminoisobutyrate	AIB	1.54(s)(6)	Juice	[25,26]
α-Glucose	α-Glc	3.41(t), 3.53(dd), 3.70(m), 3.84(m), 5.23(d)(1)	All	[2,10,20,21,22,23,24,33,34,35]
α-Hydroxyisobutyric acid	HIB	1.35(s)(6)	Pericarp	[25,26]
α-Ketoglutaric acid	KG	2.46(t)(2), 3.04(m)	All	[20,25,26]
β-Alanine	β-Ala	2.51(t)(2), 3.23(t)	Pericarp	[25,26]
β-Glucose	β-Glc	3.23(dd), 3.39(t), 3.45(dd), 3.47(t), 3.72(m), 3.88(dd), 4.64(d)(1)	All	[2,10,20,21,22,23,24,33,34,35]
γ-Aminobutyric acid	GABA	1.91(m), 2.30(t), 3.02(t)(2)	Pericarp	[2,10,21,22,24,33,34]
γ-Terpinene	TPE	2.20(m), 2.32(s), 7.15(d)(1), 7.19(m)	Pericarp	[30,31]

^a^ Multiplicity: s, singlet; d, doublet; t, triplet; q, quartet; dd, doublet; m, multiplet; AB, AB spectrum. ^b^ Underlined peaks indicate the characteristic signals of each component in the citrus juice and peel for the quantitative analysis. The numbers in the parentheses indicate the proton numbers contributing to the characteristic NMR signals. ^c^ All—both in the juices and pericarps.

## Data Availability

The data used in the current study are available from the corresponding author on reasonable request.

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
