# Peer review of "Compositional Analysis of Four Kinds of Citrus Fruits with an NMR-Based Method for Understanding Nutritional Value and Rational Utilization: From Pericarp to Juice"

_molecules, 2022, doi:10.3390/molecules27082579_

Round 1
Reviewer 1 Report
Title: NMR-based quantitative analyses of four kinds of citrus fruits for nutritional understanding and comprehensive utilization: From pericarp to juice
Molecules
Manuscript Number: molecules-1640355
The manuscript applies the NMR technology combined with multivariate statistical analyses to comparatively analyze the compositional differences of four kinds of citrus species including lemon, dekopon, sweet orange and pomelo, and to quantify the nutritional components in the pericarp and juice. In my opinion, it can add new insights to the field. However, it is not satisfactory in this current form. There are many issues that should be corrected and clarified before it can be accepted for publication.
The title must be revised to cover the work done. It needs total rewriting. Many words choices are not correct. NMR-based quantitative analyses???
The nutritional composition is not really done in the study. Please revise it throughout the manuscript. It might change to chemical profile
The English language must be improved. The manuscript needs extensive and critical revision.
Examples
Line 37-38… due to its favorite flavor, aroma and taste, high nutritional value and numerous health benefits.
Line 42… antioxidant, anti-aging, anti-cancer, antimicrobial, and health protective properties [2-4]. Change antimicrobial to antimicrobe or put activities at the end
Lines 51-52. A large amount of citrus peel is produced around the world each year, therefore how to deal with it has become a challenging problem [7].. please rewrite the sentence.
Lines 81-82. NMR can simultaneously accurately identify and quantify the multiple chemical components with high selectivity, so there is no need for complicated chemical pretreatment [16]. Please revise the sentence
The methods should be well described. Please cite the reference used in all methods and clearly describe the methods
Four species of citrus fruits were purchased from their main geographical origins in China. Among them, 10 sweet orange (Citrus sinensis) samples were from Meishan City, Sichuan Province, 10 lemon (Citrus limon) samples were from Ziyang City, Sichuan Province, 10 pomelo (Citrus maxima) samples were from Luzhou City, Sichuan Province, and 10 dekopon (Citrus reticulate Siranui) samples were from Chengdu City, Sichuan Province of China. Why are the samples collected from different locations??? The variation should be considered
Lines 133-135. The pericarp slurry was homogenized with 1 mL of methanol, 2 mL of chloroform, and 1 ml of distilled water and votexed for 60 s. After 10 min partition on ice, the mixture was centrifuged for 10 min at 10,000 g at 4 ºC. The supernatant was transferred into a 1.5-mL EP tube and lyophilized for 24 h to remove methanol and water??? How about chloroform
Why for juice deuterated phosphate buffer solution was (200 mM, pH=3.82) and for pericarp 300 mM, pH=6.5
Please mention the parameters for 1D NMR and did you run presaturation to supress the water peak. Do you run 2D NMR to confirm the identity of the assigned metabolites?
The statistical analysis to show significance differences is not reported in the Tables and Figures. Please provide that and show the significance in the result
Please present 2D NMR results to confirm the identity of the assigned metabolites
Also, the LC-MS data is needed to confirm the secondary metabolites. Please present if it is available or provide the reference for each metabolite
Section 2.3. Univariate and multivariate statistical analyses is too long and present unnecessary information. Please revise it. For the scaling method, just explain the one you used in the study. for NMR, Pareto is the most useful method.
Please describe how you do the quantification. It should be relative to TSP
Line 193… Finally, a four-dimensional volcano map was used to identify the potential biomarkers, which play a great role in the separation of the pair-wise groups. Please change biomarkers to chemical markers and consider the use of each one of them
Please revise title of section 3.2. Understanding the compositional differences between the different species??? of citrus juices
The results are not well organized, and discussion should be improved. Many results are presented and are not well discussed
Figure 1 is not clear. please present the identified metabolites. The figure should be self-explanatory
In Figure 2, Why the PLS-DA models are presented. We can see not much differences in the separation in PCA compared to PLS-DA. The loading plot that shows the compounds contributing to the separation is not shown and no validation of PLS-DA are presented. Please revise accordingly
In Figure 3 and 4, why OPLS-DA is presented? What is the purpose of presenting it? The OPLS-DA can be done for the separation of 2 groups which we can not see it in those models. For me it is wrong. The compound contributing to the separations can be concluded from the previous models and OPLS-DA should be removed.
The conclusion should be revised. The objective is not answered. Please report the metabolite differences in each variety
Reviewer 2 Report
The manuscript is very interesting, since the NMR technique is very noble and provides a lot of information. Mainly the good thing about the NMR technique is that it does not alter the analyzed sample, so the information obtained is directly from the sample.
The conclusions would have to be more forceful, not simply a reading of the spectra, which is routine in the NMR technique. In addition, the bibliography is outdated, only 30% of the references are from the last 5 years.
Reviewer 3 Report
The ms. “NMR-based quantitative analyses of four kinds of citrus fruits for nutritional understanding and comprehensive utilization: From pericarp to juice” (Ms. Ref. No. molecules-1640355-v1) presents original results on the metabolites profiling of sweet orange, lemon, pomelo and dekopon by means of quantitative NMR experiments.
The topic falls within the aims and scopes of the Molecules journal. Originality is fair.
There is a lot of work involved and the ms. has reasonable merit. However, there are important issues that negatively affect the quality of the ms.
Major issues:
- The main drawback of the manuscript is that the obtained NMR data were not confirmed through a different method. In this case, it is indicated that the NMR data – both qualitative (e. species identification) and quantitative – be compared with chromatographic data (GC-MS or GC with etalon standards), or verified through NMR with etalon compounds. Please supply with the metabolites profiling through GC. This would be the most convenient approach. Alternatively, if you do not have access to GC or have time constraints for the revision, please at least provide with suitable references for each identified species in Table 1 (as an additional column), for confirmation. For most of the reported species, you can find and compare the marker resonances and the corresponding chemical shifts and signal allure (s, d, t, dd, m) in the following works (or any other relevant paper):
- 1H NMR Reveals Dynamic Changes of Primary Metabolites in Purple Passion Fruit (Passiflora edulisSims) Juice during Maturation and Ripening (Agriculture, 2022, DOI: 10.3390/agriculture12020156);
- A multi-faceted comparison of phytochemicals in seven citrus peels and improvement of chemical composition and antioxidant activity by steaming (LWT, 160, 2022, 113297, DOI: 10.1016/j.lwt.2022.113297; I consider this paper very relevant for you, because it reports several of the species you have also identified, but through chromatographic methods, which may confirm your results);
- 1H-NMR Metabolomics as a Tool for Winemaking Monitoring, Molecules 2021, 26, 6771. https://doi.org/10.3390/molecules26226771; many of the discussed species have also been identified in grape juice and wine. This paper is also interesting on how to report NMR data.
- Understanding metabolic perturbations in palm wine during storage using multi-platform metabolomics (LWT, 155, 2022, article no. 112889, DOI: https://doi.org/10.1016/j.lwt.2021.112889) – This is also very relevant, because it reports several of the compounds found in citrus extracts, also identified and determined through 1H-NMR spectroscopy. Complementary information for compounds confirmation may be found in: Influence of common and selected yeasts on wine composition studied using 1H-NMR spectroscopy, Revista de Chimie, 2011, and in Composition changes in wines produced by different growing techniques examined through 1H-NMR spectroscopy, Revista de Chimie, 2011. It is indicated to include in Table 1 references for each metabolite, based on both NMR and GC data (where available). If you find multiple references for a certain compound, then it is a solid proof for the accuracy of the identification.
- Please provide the algorithm for the quantitative determination of the reported species. I suppose TSP was used as an internal standard not only as a reference for the 0 ppm calibration , but it may also be used as a reference for the quantitative measurement, if added in precise amounts.
- In the M&M section: How many replicates were considered for each sample? How many extractions? How many spectra recorded?
- Regarding Table 1: apart from the additional column with references for each identified compound (see comment #1), I suggest adding the number of equivalent protons and the integral values. This will also help in understanding the algorithm for the quantitative determination (see comment #3). Please revise.
- Also regarding Table 1: I am intrigued that the authors report that NMR resonances appear at different chemical shifts when found in juice compared to pericarp (for example, but not limited to, in the case of citric acid). Why? An explanation is necessary.
- Regarding Table 1: 4-Nitrophenol seems odd to be found in fruit juices. Therefore, I would not expect to find papers reporting it in fruit extracts. However, its NMR resonances have been reported and may be compared with those reported in the current submission, for confirmation (Synthesis of a new substrate for exo-galactofuranosidases, 2-hydroxy-4-nitrobenzene 1-yl β-d-galactofuranoside).
- In addition, it is odd to find lactose in a citrus fruit. A reference is necessary to support this finding.
Minor issues:
- Line 126: “was put on the table for 5 min at room temperature” does not sound well. Instead, I suggest “was allowed to stand for 5 min at room temperature”.
- Please revise the References section and make sure all the identification elements (e.g. pages) are provided. Please ensure compliance with journal’s format for the references.
- Please revise English throughout the document.
Given the completed score sheet and the comments above, after careful evaluation, the ms. “NMR-based quantitative analyses of four kinds of citrus fruits for nutritional understanding and comprehensive utilization: From pericarp to juice” (Ms. Ref. No. molecules-1640355-v1) needs Major Revision according to comments before being considered for publication in Molecules journal.
Round 2
Reviewer 1 Report
Dear authors. In my opinion the manuscript has improved significantly. however, there are few comments that they should be taken into consideration.
Please revise the English further especially for the added text. Line 159..... methanol signals (δ3.37-3.35 for citrus pericarp samples). please revise the range from lowest to highest signal
line 257.... which gives a sour and astringent tast of lemon juice. please revise the sentence and change tast to taste
In supplementary material, the legend are not clear. also authors did not mention what are w, NQ, and /. not statistical analysis was performed
b Not available for the quantitative analysis due to the low signal-to-noise ratio (<10)
c The quantitative results are given in the form of mean ± standard deviation, which are calculated from ten samples in the each group.
d Not detectable in this variety of citrus juice or citrus pericarp
e Not available for the quantitative analysis due to the interference signal of this component by the water peak
Please present them in suitable and clear form and the same for table S2
Reviewer 3 Report
Comments on the ms. “Compositional analysis of four kinds of citrus fruits by NMR-based method for understanding nutritional value and rational utilization: From pericarp to juice " (revised title, Ms. Ref. No. molecules-1640355- v2):
After the first review round, the authors have suitably addressed the reviewers’ comments. Proper changes have been made in the ms. according to suggestions and consequently, the ms. was considerably improved compared to its initial submission. I agree with the modifications on the manuscript. However, there are still a couple of minor improvements that I would like to point out.
- Regarding comment #5: I agree with the authors’ explanation. I think this explanation is worth being added into the manuscript, because the data displayed in tables should also be commented. It should not remain only as a confidential comment to reviewers and the editors. Please revise.
- Regarding comment #1: I agree with your response and I think that you should mention that the marker resonances were also established/confirmed using in house experiments on etalon compounds.
- Regarding comment #7: As I said previously, data reported in tables should also be commented. I have seed that other authors have also reported lactose in fruit juices. And you have also found malic acid (as expected) into the studied matrices. In these conditions, I wonder if you might have disregarded the lactic acid. Please check the resonances at ~1.37 ppm (see Todasca et al., Composition Changes in Wines Produced by Different Growing Techniques Examined Through 1 H-NMR Spectroscopy, Revista de Chimie, 62(2), 2011, pp. 131-134). It is highly probable to occur via the malo-lactic fermentation. Please revise and comment accordingly. Please also check the markers of the rest of the compounds against those from the previous article also with respect to comment #5, because there is a pH change during fermentation.
Consequently, after careful examination of the revised ms. “Compositional analysis of four kinds of citrus fruits by NMR-based method for understanding nutritional value and rational utilization: From pericarp to juice " (revised title, Ms. Ref. No. molecules-1640355- v2), my recommendation term is MINOR REVISION.
